# Microglia-Based Gene Expression Signature Highly Associated with Prognosis in Low-Grade Glioma

**DOI:** 10.3390/cancers14194802

**Published:** 2022-09-30

**Authors:** Evelien Schaafsma, Chongming Jiang, Thinh Nguyen, Kenneth Zhu, Chao Cheng

**Affiliations:** 1Department of Microbiology and Immunology, Dartmouth College, Hanover, NH 03755, USA; 2Department of Medicine, Baylor College of Medicine, Houston, TX 77054, USA; 3Medical School, UT Southwestern Medical Center, Dallas, TX 77054, USA; 4The Institute for Clinical and Translational Research, Baylor College of Medicine, Houston, TX 77054, USA; 5L. Duncan Comprehensive Cancer Center, Baylor College of Medicine, Houston, TX 77054, USA; 6Department of Biomedical Data Science, Geisel School of Medicine at Dartmouth, Lebanon, NH 03755, USA

**Keywords:** microglia, glioma, glioblastoma, prognosis

## Abstract

**Simple Summary:**

Gliomas make up ~80% of malignant brain tumors in adults and are responsible for the majority of deaths from primary brain tumors. Consequently, a better understanding of the malignant features of the TME in glioma is pertinent. The aim of our study was to evaluate the expression of immune-related genes (IRGs) in glioma and their association with patient prognosis. We utilized several approaches to interrogate the glioma immune microenvironment. We found that immune genes are generally negatively associated with survival and that overall survival was significantly lower in those with a high level of microglia infiltration. The microglia abundance was significantly associated with common genomic aberrations. Lastly, we generated a 23-gene expression signature that is highly associated with patient prognosis, independent of clinical variables. These findings are relevant to investigators in the glioma field, those working in biomarker development, but also to individuals working on glioma therapeutics.

**Abstract:**

Gliomas make up ~80% of malignant brain tumors in adults and are responsible for the majority of deaths from primary brain tumors. The glioma tumor microenvironment (TME) is a dynamic, heterogeneous mixture of extracellular matrix and malignant and non-malignant cells. Several ongoing clinical trials are evaluating the efficacy of therapies that target non-malignant cells, particularly immune cells. Consequently, a better understanding of the TME in glioma is pertinent. We utilized several gene expression datasets to evaluate the relationship between immune-related genes (IRGs) and patient prognosis. We generated microglia signatures using single-cell RNAseq data from human and mouse glioma cells to infer microglia abundance. Lastly, we built a LASSO Cox regression model that predicts patient survival. We found that 428 IRGs were negatively associated with survival in glioma patients. Overall survival was significantly lower in those with a high level of microglia infiltration. In addition, we also found that microglia abundance was significantly associated with several common genomic aberrations, including *IDH2* and *TP53* mutations. Furthermore, we found that patients with high risk scores had significantly worse overall survival than those with low risk scores in several independent datasets. Altogether, we characterized immune features predictive of overall survival in glioma and found that microglia abundance is negatively associated with survival. We developed a 23-gene risk score that can significantly stratify patients into low- and high-risk categories.

## 1. Introduction

Gliomas make up ~80% of malignant brain tumors in adults and are responsible for the majority of deaths from primary brain tumors [1]. Gliomas can originate from different types of glial cells, including astrocytes, oligodendrocytes and ependymal cells. As these cells are present throughout the nervous system, gliomas can appear in various parts of the brain and spinal cord. The 2021 World Health Organization Classification of Tumors of the Central Nervous System (WHO CNS) provides detailed tumor classification guidelines based on histology and immunohistochemistry [2]. Four grades of glioma are distinguished: grades I and II are considered low-grade gliomas (LGGs), and grades III and IV are high-grade gliomas (HGGs). In addition, grade IV gliomas are often designated as glioblastoma (GBM). Treatment is based on molecular profiling and tumor grade and generally involves surgery, adjuvant therapy and radiotherapy [3].

The glioma tumor microenvironment (TME) is a dynamic, heterogeneous mixture of extracellular matrix and malignant and non-malignant cells [4]. Half or more of the cells within the glioma TME are typically non-neoplastic and include neurons, glia, leukocytes and endothelial cells [5]. Valuable insights have been gained from both the study of T-cell biology in brain tumors [6,7] and the investigation of other cell populations contributing to immune suppression in the glioma TME [8,9,10]. The modulation of these cell populations in the glioma TME could improve the efficacy of immunotherapy against brain malignancies [11,12]. 

Immune checkpoint inhibitors, which mainly target checkpoint proteins, including PD-1 and anti-PD-L1, on T cells, have not yet significantly improved overall survival for patients with glioma. Examples of evaluated antibodies include Nivolumab, an anti-PD-1 antibody, Pembrolizumab, an anti-PD-1 antibody, and Ipilimumab, an anti-CTLA4 antibody. In a phase II clinical trial in resectable GBM, neoadjuvant Nivolumab (anti-PD-1) prior to tumor resection did not substantiate obvious clinical benefit [13]. Another phase II clinical trial compared Pembrolizumab (anti-PD-1) to Pembrolizumab with Bevacizumab (anti-VEGF) and found that Pembrolizumab alone or with Bevacizumab was well-tolerated but of limited benefit [14]. In a phase III clinical trial comparing Nivolumab (anti-PD-1) to Bevacizumab (anti-VEGF) for recurrent GBM, both treatments conferred similar median overall survival. Lastly, a phase I clinical trial showed that combination therapy of Nivolumab (anti-PD-1) and Ipilimumab (anti-CTLA-4) in patients with recurrent GBM did not improve overall survival but worsened treatment-related adverse events (AEs) [15]. Multiple clinical trials are still underway in glioma and GBM to evaluate, for example, the combination of immune checkpoint inhibitors and radiotherapy [16]. Given the extraordinary success of immune checkpoint inhibitors in other cancer types, including metastatic melanoma or lung cancer, the hope of successfully treating glioma patients with immune checkpoint inhibitors remains.

Consequently, a better understanding of the TME in glioma is pertinent. In this study, we evaluated the expression of immune-related genes (IRGs) in glioma and their association with patient prognosis. We characterized immune features predictive of overall survival and developed a 23-gene risk score that can stratify patients as having a low and high risk of death due to glioma. An overview of our study can be found in Appendix A.

## 2. Materials and Methods

### 2.1. Data Utilized in this Study

Level 3 TCGA RNAseq data and clinical information for low-grade glioma (LGG, *n* = 515) and glioblastoma multiforme (GBM, *n* = 160) were obtained from TCGA on FireBrowse (gdac.broadinstitute.org/ (accessed on 29 July 2022)). TCGA MAF files for gene mutation analyses were obtained from https://gdc.cancer.gov/about-data/publications/pancanatlas (accessed on 29 July 2022). All genes in which non-silent mutations occurred were considered to be mutated. Total mutation burden (TMB) was represented as the sum of all non-silent mutations in a given TCGA sample. The copy number variation (CNV) data were also downloaded from Firehose, which provided DNA segments deviated from normal copy numbers (copy number = 2) in each tumor sample. The CNV burden of a sample was calculated as the total size (in bp) of genomic regions covered by those segments. Macrophage regulation scores, leukocyte and lymphocyte infiltration scores and IFNγ response and TGFβ response scores for TCGA samples were downloaded as a supplemental file from prior work [17]. We note here that the TCGA classification of low-grade gliomas includes grades II and III, although grade III gliomas are considered high-grade gliomas clinically. The GBM dataset contains all grade IV gliomas. In addition, TCGA glioma subtypes were assigned based on the fourth WHO CNS version and may not reflect classification based on the latest WHO CNS version [2]. The transcriptomic and clinical data of glioma patients from the Rembrandt data (*n* = 580) [18] were accessed through the Rembrandt data portal (https://sites.google.com/georgetown.edu/g-doc/home (accessed on 29 July 2022)). The transcriptomic and clinical data of glioma patients from the Chinese Glioma Genome Atlas (CGGA) databases (mRNAseq_693, *n* = 693) were downloaded from cgga.org.cn (accessed on 29 July 2022).

### 2.2. Curation of Immune-Related Genes (IRGs)

Immune-related genes (IRGs) were obtained from Supplementary Table S6 from Charoentong et al. [19], Supplementary Table S1 from Bindea et al. [20] and Supplementary Table S1A from Xu et al. [21]. All genes from immune cells were collected; i.e., marker genes attributed to cancer cells were excluded and combined into a single list of IRGs totaling 831 genes.

### 2.3. Immune Cell Inference

Immune infiltration scores of six immune cells were calculated using Binding Association with Sorted Expression (BASE) [22], a rank-based gene set enrichment method. Immune cell infiltration using this method has been detailed and validated in previous publications [23,24]. Briefly, BASE uses immune-cell-specific weight profiles and patient gene expression data to infer immune cell infiltration for each patient and immune cell type. Full details on the calculation and validation of the immune infiltration scores can be found in [23,24]. Similarly, BASE was used to calculate single-cell-based microglia scores using microglia signatures (see next section). 

### 2.4. Generation of Microglia Signatures

Glioma single-cell-RNAseq datasets from human [25] and mouse gliomas [26] were obtained from previous publications. Cluster annotations were obtained from these works, and the biological interpretation of each cluster is extensively described in the corresponding publications. For each human/mouse cluster, a list of marker genes was provided by identifying genes that were overexpressed in the corresponding cluster as compared to all other clusters. These cluster-specific marker gene sets were used as microglia signatures [27]. In total, 9 human and 20 mouse microglia signatures were defined. Given an LGG/GBM gene expression dataset, the BASE algorithm was used to calculate sample-specific microglia scores for each signature. For all mouse microglia signature genes, we queried for homologous human genes using biomaRt [28] and used these homologs as signature genes. Of note, microglia signatures were represented as gene sets without assigning weights to genes. In this case, the BASE algorithm degenerated into a method like the single-sample GSEA analysis [29]. A higher microglia score indicates that the corresponding subtype of microglia cells is more abundant in the tumor. 

### 2.5. Lasso Cox Regression

The TCGA LGG dataset was randomly divided into a training and testing set with a 1:1 ratio. The training set was analyzed to identify potential prognostic genes, and both the testing set and the entire set were used for validation. First, univariate Cox proportional hazards regression analysis was used to evaluate the association between the expression of the 831 IRGs and overall survival. Genes with a *p*-value of <0.05 based on the log-rank test were selected as candidate genes. Second, least absolute shrinkage and selection operator (LASSO) Cox regression analysis from the R glmnet package was employed to screen the IRGs most associated with overall survival in a multivariate model, which resulted in 23 genes (*ADAMTSL2*, *APOBEC3C*, *ARHGAP12*, *CARHSP1*, *CBFB*, *CD274*, *EVL*, *FAM161A*, *HIST2H2BE*, *HSPB1*, *ITGAV*, *MYBL1*, *PLAT*, *RCCD1*, *SAR1A*, *SLC25A45*, *SMC4*, *STAP1*, *TFDP1*, *TGIF1*, *TMEM55B*, *TNFRSF11B*, *WEE1*). These 23 genes composed the final risk score, which is described as follows:$$iskscore=\overset{n}{\underset{i=0}{\sum }}\beta_{i}x_{i}$$
where $\beta_{i}$ refers to the coefficients of each gene and $x_{i}$ represents the expression value of the gene. 

### 2.6. Survival Analysis

For univariate and multivariate survival analyses, Cox proportional hazards models were calculated using the “coxph” function from the R “survival” package. Survival curves were visualized using Kaplan–Meier curves using the “survfit” function from the R “survival” package. Median immune cell infiltration scores were used to stratify patients into “high” and “low” groups for univariate analyses. For multivariate analyses, an infiltration score of 0 was used as a separator to stratify patients into “high” and “low” groups. Differences in survival distributions in each Kaplan–Meier plot were calculated using a log-rank test using the “survdiff” function from the R “survival” package.

### 2.7. Statistical Analyses

The Spearman correlation coefficient (SCC) was reported for all correlation analyses as the assumptions underlying the Pearson correlation (i.e., normal distribution, homoscedasticity or linearity) were not met. The SCC was calculated using the R function cor, and significance was assessed using cor.test. Principal component analysis (PCA) was performed using the prcomp R function. Principal component coordinates for each sample were extracted using the factoextra R package (https://github.com/kassambara/factoextra (accessed on 29 July 2022)). Principal component 1 (PC1) was used to represent microglia infiltration. The sensitivity and specificity of the diagnostic and prognostic prediction models were analyzed by the ROC curve and quantified based on the area under the ROC curve (AUC). All statistical tests were two-sided, and *p*-values < 0.05 were considered statistically significant. All statistical analyses were performed using R software (Version 3.5.2).

### 2.8. Data Availability

All data available in this study are publicly available. These data can be found at: gdac.broadinstitute.org/ (accessed on 29 July 2022), https://gdc.cancer.gov/about-data/publications/pancanatlas (accessed on 29 July 2022), https://sites.google.com/georgetown.edu/g-doc/home (accessed on 29 July 2022), http://www.cgga.org.cn/ (accessed on 29 July 2022).

## 3. Results

### 3.1. Immune-Related Genes (IRGs) Are Negatively Associated with Prognosis in Glioma

The glioma TME is heterogeneous and incorporates many different cell types. To focus in on infiltrating immune cells, we compiled a list of 831 immune-related genes (IRGs) from several publications [19,20,21] and evaluated the expression of these genes in LGGs and GBM. We observed that the majority of IRGs were negatively associated with survival in both LGGs and GBM (Figure 1A,B). This was especially prominent in LGGs, where 428 of the 520 significant IRGs were negatively associated with survival, indicating that higher expression of these IRGs conferred shorter survival. Comparing the expression of IRGs between LGGs and GBM, we found that GBM expressed the highest levels of IRGs, with 294 IRGs overexpressed in GBM and only 71 IRGs overexpressed in LGGs (Figure 1C). Several immune checkpoints were among the IRGs associated with poor survival in LGGs, including *HAVCR2* (log-rank *p* = 0.001), *CD274* (log-rank *p* = 0.001), *CD276* (log-rank *p* = 4 × 10^–5^) and *CTLA-4* (log-rank *p* = 0.005) (Figure 1D). Thus, IRGs are overexpressed in LGGs and GBM tumors with poor prognosis, and these include immune checkpoint molecules. Since many more IRGs were differentially expressed in LGGs as compared to GBM, we focused the remainder of the study on LGGs.

Several types of immune cells infiltrate into the glioma TME. For example, T cells, tumor-associated macrophages (TAMs)/monocytes, NK cells, B cells, neutrophils and dendritic cells (DCs) have all been observed in glioma tumors [10]. To further elucidate which immune cell types confer a poor prognosis in glioma, we estimated the abundance of six immune cell types commonly present in the TME: memory B cells, naïve B cells, CD4+ T cells, CD8+ T cells, NK cells and monocytes. We then clustered patients based on their immune infiltration profile and observed two groups of patients: patients with high infiltration of monocytes but no other immune cell types (group 1) and patients with high infiltration of all immune cell types except monocytes (group 2) (Figure 2A). When comparing overall survival, patients with high monocyte infiltration (group 1) had significantly shorter survival as compared to patients with a broader immune cell infiltration pattern (group 2) (Figure 2B, log-rank *p* = 9 × 10^–5^). 

We observed a similar pattern when investigating the relationship between overall survival and individual immune cell types; higher infiltration of naïve B cells (log-rank *p* = 0.001), memory B cells (log-rank *p* = 8 × 10^–4^) and CD4+ T cells (log-rank *p* = 0.001) was associated with longer overall survival, whereas higher infiltration of monocytes was associated with shorter overall survival in LGGs (log-rank *p* = 6 × 10^–4^) (Figure 2C). We confirmed the significant association of naïve B cells and monocytes with survival in two independent datasets (Appendix A). Although CD8+ T cell infiltration was not significantly associated with survival in the TCGA dataset, the two independent LGG datasets both showed a significant association between higher CD8+ T cell infiltration and longer overall survival (Appendix A). Lastly, higher infiltration of both naïve and memory B cells was significantly associated with longer survival in GBM (Appendix A). Thus, immune cell infiltration is associated with prognosis in glioma. In LGGs, high monocyte infiltration is associated with shorter survival, and the infiltration of other immune cells is associated with a better prognosis. 

### 3.2. Microglia Abundance Is Negatively Associated with Prognosis in Glioma

Microglia are the resident macrophages of the brain [1], and the above infiltration of monocytes likely resembles, at least partially, the infiltration of microglia into the tumor. Indeed, we observed that several genes prominently expressed in microglia, *CCR5* and *TREM2*, were also negatively associated with patient survival (Figure 3A), mirroring the negative association between monocytes and overall survival (Figure 2C). We thus sought to improve the inferred monocyte profile to resemble microglia infiltration more closely. We obtained two single-cell RNA-seq (scRNA-seq) datasets, one from human [25] and one from murine gliomas [26], and isolated all cell clusters that were designated as microglia cells by the study authors. This resulted in 9 clusters for human and 20 clusters for mouse microglia cells. For each microglia cluster, we generated a gene expression signature (see Methods) and inferred the abundance of these signatures in LGG patients. 

We observed that almost all microglia signatures were strongly associated with monocyte infiltration (Appendix A), again suggesting that our monocyte signature likely captures signals from infiltrating microglia cells. In addition, inferred microglia abundance was highly positively correlated with several markers expressed in microglia cells, including ITGAM, CCR5 and FCGR2A (Appendix A). Lastly, the majority of IRGs (Appendix A) and immune checkpoint genes (Appendix A) were positively associated with microglia signature scores. It thus seems that our inferred microglia scores represent intratumoral microglia abundance and are positively associated with IRG expression, including checkpoint genes.

We next assessed the relationship between microglia infiltration and overall survival. Almost all 29 signatures, except for 1 human and 3 mouse signatures, were negatively associated with overall survival (Figure 3B). For example, higher inferred infiltration of hC1_MG1, resembling microglia with homeostatic functions [25], and hC9_MG9, resembling proliferating microglial cells [25], was associated with shorter overall survival (Figure 3C). In addition, higher abundance of mC3_Ube2c, resembling proliferating microglial cells [26], and mC9_center, resembling microglia with interferon expression (26), was also associated with shorter overall survival (Figure 3C). When correlating the inferred levels of all signatures with each other, we noticed that all signatures were highly positively correlated with one another (Figure 3D), suggesting high overlap in the signals captured by these different signatures. We thus aimed to unify these signatures and generate a single signature that resembles all microglia signatures. We performed dimensionality reduction (principle component analysis (PCA)) on the inferred human microglia signature scores and observed that the first principle component (PC) was highly positively correlated with all microglia signatures (Appendix A) and captured 72% of the variation among patients (Figure 3E). This suggested that PC1 can represent all human microglia signatures, and we thus utilized PC1 to represent overall microglia infiltration. When comparing overall survival between patients with high microglia abundance (PC1-high) and low microglia abundance (PC1-low), we observed that patients with high microglia abundance had significantly shorter survival (log-rank *p* = 4 × 10^–5^, Figure 3F). In conclusion, high microglial infiltration can be captured by gene-expression-based microglia abundance inference and is associated with shorter survival.

We next sought to validate our findings in the TCGA-LGG dataset in independent gene expression datasets. Since both independents datasets are based on human glioma biopsies, we focused on the 9 human microglia signatures. We indeed confirmed that almost all human signatures were significantly associated with overall survival in the Rembrandt and CGGA datasets and that PC1-high patients had significantly shorter survival (Figure 4A–D), confirming our earlier findings. Several subtypes of glioma exist, with astrocytoma, oligoastrocytoma and oligodendroglioma being the most common subtypes [1]. As each subtype originates from a different cell type and consequently confers different morphologies, we investigated what would occur if microglia abundance were different as well. We indeed observed a difference among subtypes: astrocytomas contained the highest levels of microglia infiltration, whereas oligodendrogliomas had the lowest levels of microglia infiltration (Figure 4E). When assessing overall survival, no difference in overall survival in astrocytomas was observed when comparing high vs. low microglia infiltration, but patients with lower microglia infiltration had significantly longer survival than patients with oligoastrocytoma and oligodendroglioma (Figure 4F). Lastly, we evaluated the relationship between overall survival and microglial abundance while adjusting for several important clinical variables known to be associated with prognosis. We observed that microglia infiltration was still associated with prognosis in multivariate regression analysis (Figure 4G), suggesting that microglia abundance is an independent prognosis marker. In conclusion, microglia infiltration is associated with survival, varies among glioma subtypes and can be used as a prognostic marker independent of clinical variables.

### 3.3. Microglia Infiltration Is Associated with the mRNA Expression of Immune Checkpoint Genes and Immune Regulatory Pathways

The glioma TME is often highly infiltrated by non-neoplastic cells, with half or more of the cells within the glioma TME being non-neoplastic [5]. We thus wanted to evaluate the relationship between microglial abundance and non-neoplastic cells, particularly immune cells, in more detail. We first correlated microglial abundance with our curated IRGs and observed that a substantial fraction of IRGs was positively associated with microglial abundance, including immune checkpoint genes such as C10orf54 (VISTA), HAVCR2 (TIM-3) and CD274 (PD-L1) (Figure 5A). In addition, genes expressed predominantly in microglia were strongly positively associated with microglial abundance (Figure 5B), including CD68 and MSR1 (Figure 5C), suggesting again that PC1 represents microglial infiltration. Similarly, a macrophage regulation score was also positively associated with microglia abundance (Figure 5D). This score represents a colony-stimulating factor-1 (CSF1) response [30], which is the primary regulator of tissue macrophages and induces proliferation, differentiation and survival of macrophages and microglia [31]. Lastly, microglia abundance was more highly correlated with M2 macrophages (SCC = 0.50) as compared to M1 macrophages (SCC = 0.17) (Appendix A), suggesting that microglia abundance is associated with a suppressive TME.

When assessing the relationship between microglia infiltration and general immune cell infiltration, we observed a positive association with both leukocyte and lymphocyte infiltration (Figure 5E). We previously noticed a positive correlation between single-cell-based microglial signatures and B-cell infiltration (Appendix A) and hypothesize that the positive correlation observed in Figure 5E was at least in part due to B-cell infiltration. Lastly, we inferred pathway activity of several immune pathways, including IFNγ and TGFβ pathways. Both pathways were positively correlated with microglia infiltration (Figure 5F). In conclusion, microglia infiltration is associated with the expression of many immune checkpoint proteins and immune-related pathways. 

### 3.4. Microglia Infiltration Is Associated with Specific Genomic Alterations

After evaluating the relationship between immune-related characteristics and microglia infiltration, we next investigated the relationship between genomic alterations and microglia infiltration. Several genomic alterations are prominent in glioma, including isocitrate dehydrogenase (*IDH*) mutations and *CDKNA2A* deletions. We first evaluated somatic mutations and observed that several common gene mutations were associated with microglia infiltration (Figure 6A). For example, patients with *TP53* mutations tended to have higher levels of microglia in their tumor, whereas patients with *CIC* or *IDH2* mutations tended to have lower microglia infiltration (Figure 6B). A large number of copy number variations (CNVs) were associated with microglia infiltration, most notably EGFR amplifications and CDKNA2A deletions; both of these genomic abnormalities were associated with increased microglia infiltration (Figure 6C). Lastly, neither overall mutation counts nor overall CNVs were not associated with microglia infiltration (Figure 6D,E). These results suggest that specific genomic alterations are involved in microglia infiltration and not the general level of genomic abnormalities.

### 3.5. A 23-Gene Risk Score Is Highly Associated with Overall Survival in Glioma

As we have shown previously, microglia infiltration is associated with survival independently of clinical variables (Figure 4). However, the inference of microglia abundance throughout our studies has been based on a large number of genes, each contributing to the final infiltration score. In order to generate a potentially clinically useful tool, we thus aimed to significantly reduce the number of genes in a signature that can be used to risk-stratify glioma patients. We used Lasso Cox regression for feature (gene) selection and ended up with a 23-gene risk score (Methods, Appendix A). The following genes were included in the risk score: *ADAMTSL2*, *APOBEC3C*, *ARHGAP12*, *CARHSP1*, *CBFB*, *CD274*, *EVL*, *FAM161A*, *HIST2H2BE*, *HSPB1*, *ITGAV*, *MYBL1*, *PLAT*, *RCCD1*, *SAR1A*, *SLC25A45*, *SMC4*, *STAP1*, *TFDP1*, *TGIF1*, *TMEM55B*, *TNFRSF11B* and *WEE1.* The expression of these genes was weighted and summed to generate a final risk score ($riskscore=\overset{n}{\underset{i=0}{\sum }}\beta_{i}x_{i}$ where $\beta_{i}$ refers to the coefficients of each gene and $x_{i}$ represents the expression value of the gene). This signature was significantly associated with glioma survival in the TCGA training dataset (Figure 7A) and in two independent datasets (Figure 7B,C), with higher risk scores being associated with shorter overall survival. This signature is specific to glioma and not GBM as only one of the three datasets showed a significant relationship between overall survival and the risk score in GBM (Figure 7D, Appendix A). Notably, the risk score was highly associated with survival when evaluated in a multivariate model with several clinical variables (Figure 7E). In this model, the hazard risk (HR) of our risk score was much larger than that of tumor grade, a commonly used risk stratification measure in glioma [3]. In conclusion, we developed a 23-gene risk score that is highly associated with prognosis in glioma and can be used in addition to other clinical variables.

## 4. Discussion

Glioma is a heterogeneous disease in which many non-malignant cells infiltrate into the TME. Several clinical trials are underway to assess the efficacy of targeting these non-malignant cells [5], especially immune cells. To better understand the relationship between the immune system and patient survival in gliomas, we first started out by investigating the relationship between IRG expression and patient prognosis. We found that the overexpression of many IRGs is associated with shorter overall survival. However, this is contrary to the expression of immune genes in several other cancer types; expression of IRGs is associated with better survival in, for example, melanoma and kidney cancer [19]. Notably, several immune checkpoint genes were among the IRGs negatively associated with survival. Thus, this led us to investigate the effect of immune checkpoint inhibitors as a possible mechanism for these unique effects. 

Further investigation showed that the abundance of monocytes, but not other immune cell types, was negatively associated with prognosis. As microglia are the resident macrophages of the brain [5], we reasoned that this monocyte profile is likely in part reflective of microglia abundance. We used scRNAseq-based microglia signatures to evaluate the infiltration of microglia in a more efficient manner. These signatures, as well as a compiled microglia signature, were positively associated with the majority of IRGs and immune checkpoint genes. We hypothesize that microglia express several immune checkpoint genes and in turn negatively regulate T-cell activation. In addition, microglia potentially act as antigen-presenting cells (APCs), given their ability to upregulate the expression of MHC class II and costimulatory molecules [32]. Thus, in the presence of immune checkpoint expression, T-cell activation will be attenuated, and no effective antitumor response would be initiated. Alternatively, in pathological states, microglia produce proinflammatory factors such as cytokines and chemokines [33] that promote tumor invasiveness [34]. 

Microglia abundance was associated with several genomic alterations. For example, tumors with IDH2 mutations had significantly lower infiltration of microglia as compared to tumors without IDH2 mutations. It is known that IDH wildtype gliomas have a poorer prognosis and are molecularly similar to GBM [35]. This is consistent with our finding that higher microglia infiltration is associated with shorter survival. CNVs in EGFR and CDKNA2A often occur in the absence of IDH mutations [35]. These alterations were consequently associated with higher microglia infiltration. Overall, TMB and CNV burden were not associated with microglia infiltration, suggesting that specific genomic alterations affect microglia infiltration and not the magnitude of genomic aberrations.

Lastly, we developed a 23-gene risk score that can stratify high- and low-risk patients irrespective of the predictive ability of clinical variables. This risk score was validated in independent datasets and was strongly associated with prognosis. Interestingly, prognosis could only be predicted in gliomas but not high-grade gliomas, or glioblastoma (GBM). It was also notable that a much smaller number of IRGs were significantly and differentially expressed in GBM as compared to LGGs. This is likely due to the more aggressive phenotype of GBM as compared to a glioma, which confers a high risk itself. We do note that the TCGA-LGG dataset contains includes both grades II and III, although grade III gliomas are considered high-grade gliomas clinically. Thus, this observation seems specific to high-grade grade IV gliomas. The 23-gene signature could be clinically relevant as only a small number of genes need to be measured to establish a patient-specific risk score. 

Previous studies have aimed at identifying TME-related factors associated with prognosis. For example, Ni et al. (2020) used the ESTIMATE algorithm [36] in combination with network analysis to identify a signature of 25 immune genes associated with prognosis [37]. Similar to our study, high signature scores, i.e., high immune infiltration, was associated with shorter overall survival. While this study identified a non-specific immune signature that was validated in one dataset, we developed a microglia-specific signature and validated our signature in multiple independent datasets. The increased specificity of our signature adds biological relevance and could aid in patient selection for microglia-specific interventions in the future.

Although our study provides valuable insights into the potential role of microglia in LGGs, we note a few limitations of our study. First, we used immune inference methods to infer the abundance of immune cells and microglia. While these methods have been thoroughly validated, a more precise quantification using, for example, immunohistochemistry, would complement our inference estimates. Second, the expression of immune checkpoints was based on mRNA data, which might not reflect protein levels. Lastly, our 23-gene signature needs to be validated in a prospective study to assess clinical efficacy. The potential difference between risk score quantification in FFPE or fresh glioma samples would also need to be evaluated to establish clinical utility.

## Figures and Tables

**Figure 1 cancers-14-04802-f001:**
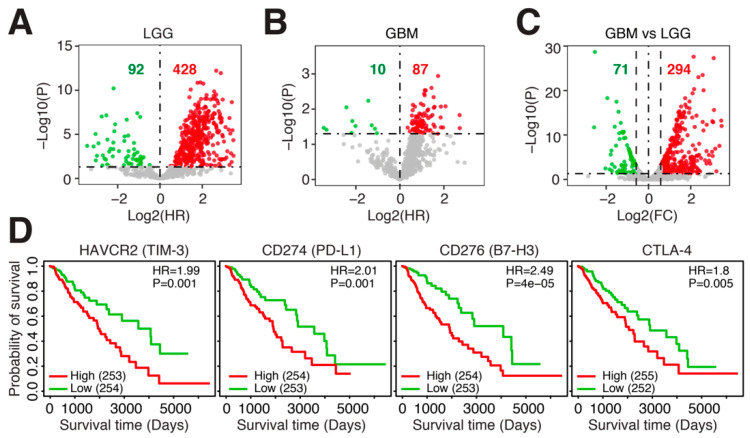
Immune-related genes (IRGs) are negatively associated with prognosis in glioma. Log2 hazard ratio of univariate Cox regression models evaluating the association between overall survival and IRG expression in (**A**). TCGA-LGG (*n* = 515) and (**B**). TCGA-GBM (*n* = 160). (**C**). Log2 fold change between IRG expression in TCGA-LGG (*n* = 515) and TCGA-GBM (*n* = 160). (**D**). Kaplan–Meier plots of four immune checkpoint genes in TCGA-LGG (*n* = 515). HR = hazard ratio. *p* = log-rank *p*-value.

**Figure 2 cancers-14-04802-f002:**
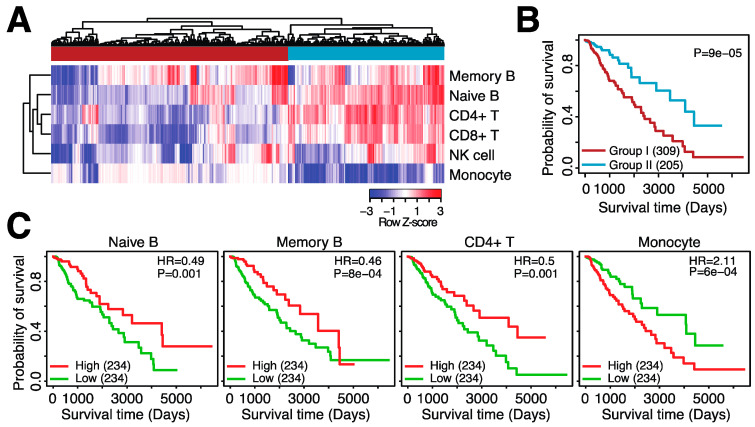
Patterns of immune infiltration in low-grade glioma. (**A**). Heatmap depicting the infiltration scores for six immune cell types in TCGA-LGG. Top sidebar indicates the grouping of the samples based on hierarchical clustering. (**B**). Kaplan–Meier plot depicting differential overall survival between group I and group II. (**C**). Kaplan–Meier plots comparing low and high infiltration of four immune cell types. HR = hazard ratio. *p* = log-rank *p*-value.

**Figure 3 cancers-14-04802-f003:**
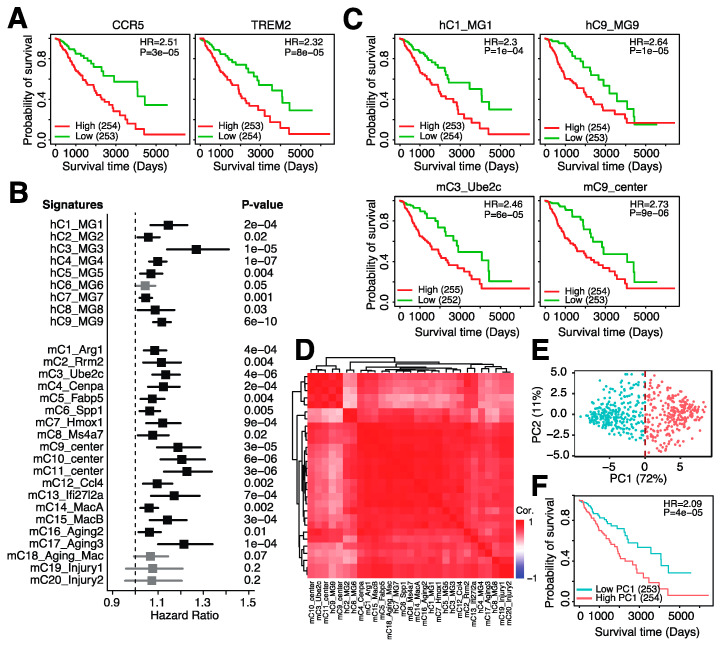
Microglia abundance is negatively associated with prognosis in low-grade gliomas. (**A**). Kaplan–Meier plot comparing overall survival between patients with low and high expression of *CCR5* (left) and *TREM2* (right). (**B**). Hazard ratios of univariate Cox regression models evaluating the association between overall survival and 9 human microglia signatures (“h”) and 20 mouse microglia signatures (“m”). (**C**). Kaplan–Meier plots depicting the association between overall survival and four microglia signatures. (**D**). Spearman correlation between the 29 microglia gene expression signatures. (**E**). Principle component analysis (PCA) on the expression of the 29 microglia gene expression signature in TCGA-LGG patients. (**F**). Kaplan–Meier plot showing the association between overall survival and principle component 1 (PC1) in TCGA-LGG.

**Figure 4 cancers-14-04802-f004:**
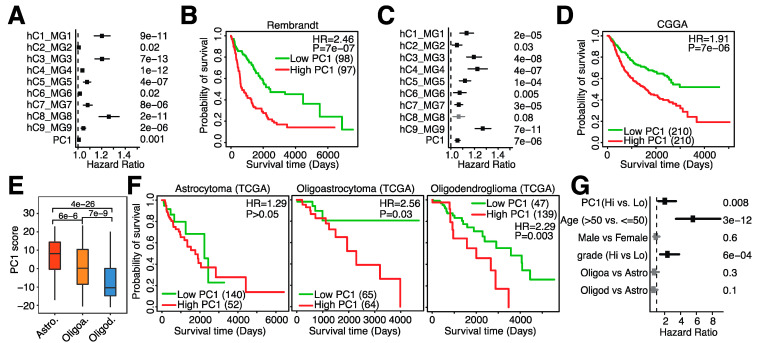
Microglia abundance is negatively associated with prognosis in independent datasets and subtypes of low-grade glioma. (**A**). Hazard ratios of univariate Cox regression models evaluating the association between overall survival and 9 human microglia signatures and PC1 in the Rembrandt dataset. (**B**). Kaplan–Meier plot depicting the association between overall survival and PC1 in the Rembrandt dataset. (**C**). Hazard ratios of univariate Cox regression models evaluating the association between overall survival and 9 human microglia signatures and PC1 in the CGGA dataset. (**D**). Kaplan–Meier plot depicting the association between overall survival and PC1 in the CGGA dataset (mRNAseq_693). (**E**). Boxplot comparing PC1 scores among glioma subtypes. Astro. = astrocytoma, Oligoa. = oligoastrocytoma and Oligod. = oligodendroglioma. (**F**). Kaplan–Meier plots depicting the association between overall survival and PC1 in different glioma subtypes in the TCGA dataset. (**G**). Forest plot depicting hazard ratios of univariate Cox regression models evaluating the association between overall survival and several clinical parameters.

**Figure 5 cancers-14-04802-f005:**
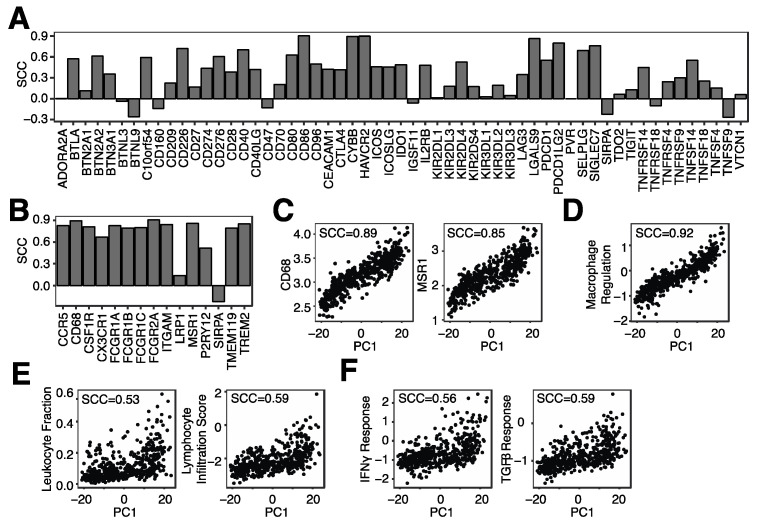
Microglia infiltration is associated with the mRNA expression of immune checkpoint genes and immune regulatory pathways. (**A**). Spearman correlation coefficient (SCC) between PC1 and the mRNA expression of several immune checkpoint genes. (**B**). SCC between PC1 and mRNA genes expressed in microglia. (**C**). SCC between PC1 and CD86 and MSR1 mRNA levels. (**D**). SCC between PC1 and a macrophage regulation score. (**E**). SCC between PC1 and leukocyte and lymphocyte infiltration. (**F**). SCC between PC1 and IFNγ response and TGFβ response.

**Figure 6 cancers-14-04802-f006:**
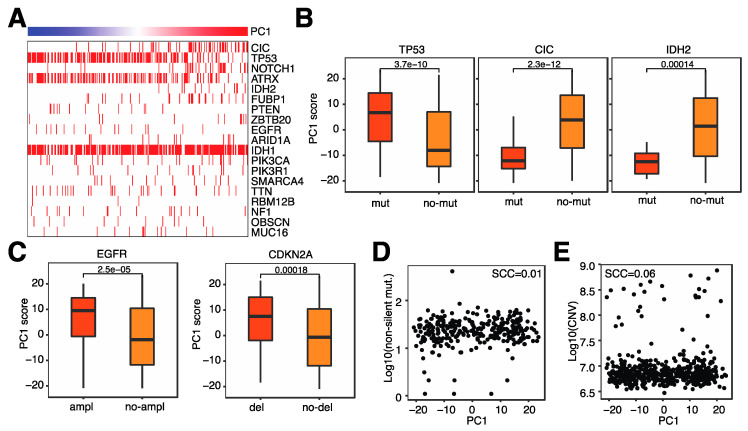
Microglia infiltration is associated with specific genomic alterations. (**A**). Heatmap indicating the incidence of coding mutations (horizontal red bars) and the level of microglia infiltration (top bar). (**B**). Box plot of microglia infiltration in patients with or without coding mutations in the specified genes. (**C**). Box plot of microglia infiltration in patients with or without copy number variations (CNVs) in the specified genes. (**D**). Relationship between microglia infiltration and the number of non-silent mutations. (**E**). Relationship between microglia infiltration and total CNVs.

**Figure 7 cancers-14-04802-f007:**
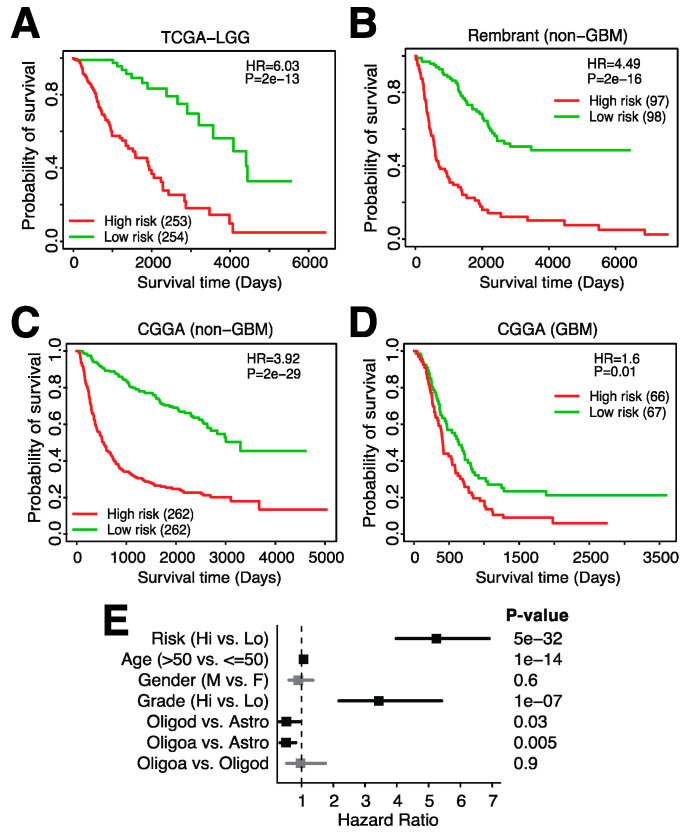
A 23-gene risk score is highly associated with overall survival in low-grade glioma. Kaplan–Meier plot showing the association between overall survival and the 23-gene risk score in (**A**). TCGA-LGG, (**B**). the Rembrandt dataset (mRNAseq_693, non-GBM), (**C**). CGGA (mRNAseq_693, non-GBM) and (**D**). CGGA (GBM). (**E**). Forest plot of hazard ratios derived from multivariable Cox regression analysis including the listed variables.

## Data Availability

Not applicable.

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
