# Peer review of "Microglia-Based Gene Expression Signature Highly Associated with Prognosis in Low-Grade Glioma"

_cancers, 2022, doi:10.3390/cancers14194802_

Round 1
Reviewer 1 Report
Dear authors,
You have done an excellent work and the only thing that I wish to point out, is that the brain tumours should be named according to the 2021 WHO Classification of Tumours of Central Nervous System. So the terms glioblastoma multiforme, oligoastrocytoma and oligodendrocytoma should be revised.
Author Response
Reviewer 1
Comment:
You have done an excellent work and the only thing I want to point out, is that the brain tumors should be named according to the 2021 WHO Classification of Tumours of Central Nervous System. So the terms glioblastoma multiforme, oligoastrocytoma and oligodendrocytoma should be revised.
Response:
We thank the reviewer for carefully reading and evaluating our manuscript. The reviewer brought up a great suggestion to update the classification according to the latest WHO Classification of Tumours of Central Nervous System version. The older WHO classification that was used in the manuscript (WHO CNS3, 2007) relied heavily on the histologic identification of cell types. Under this classification scheme, gliomas were categorized by cell type (astrocytoma, oligodendroglioma, oligoastrocytoma, or ependymoma). The latest WHO Classification of Tumours of Central Nervous System (WHO CNS5) relies more heavily on immunohistochemistry of molecular biomarkers (including for example IDH1/2 and EGFR). While attempting to update the older classification scheme, we soon discovered that information on several of these biomarkers is unavailable for most samples included in our study, making it impossible to redetermine their subtypes under WHO CNS5 or update the original subtype names into the new ones. We therefore kept the older WHO classification system but explicitly noted in the manuscript that we used an older WHO Classification of Tumours of Central Nervous System system.
In the revision, we made the following changes:
In “Method” (Lines 113-115, Page 3), we added the following sentence:
Macrophage regulation scores, leukocyte and lymphocyte infiltration scores, and IFNg response and TGFb response scores for TCGA samples were downloaded as a supplemental file from prior work (15). We note here that the TCGA classification of low-grade gliomas includes grades II and III, although grade III gliomas are considered high-grade gliomas clinically. The GBM dataset contains all grade IV gliomas. In addition, TCGA glioma subtypes were assigned based on the fourth WHO CNS version (16) and may not reflect classification based on the current sixth WHO CNS version (2). The transcriptomic and clinical data of glioma patients from the Rembrandt data(n=580)(18) were accessed through the Rembrandt data portal110(caintergator.nci.nih.gov). The transcriptomic and clinical data of glioma patients from the Chinese111Glioma Genome Atlas (CGGA) databases (mRNAseq_693, n=693) were downloaded from112cgga.org.cn.
In “Introduction” (Lines 54-56, Page 2), we added the following sentence:
Gliomas make up ~80% of malignant brain tumors in adults and are responsible for the majority of deaths from primary brain tumors (1). Gliomas can originate from different types of glial cells, including astrocytes, oligodendrocytes, and ependymal cells. As these cells are present throughout the nervous system, gliomas can appear in various parts of the brain and spinal cord. The 2021 World Health Organization Classification of Tumors of the Central Nervous System (WHO CNS) provides detailed tumor classification guidelines based on histology and immunohistochemistry (2). Four grades of glioma are distinguished: grades I and II are considered low-grade gliomas (LGG) and grades III and IV are high-grade gliomas (HGG). In addition, grade IV gliomas are often designated as glioblastoma (GBM). Treatment is based on molecular profiling and tumor grade and generally involves surgery, adjuvant therapy and radiotherapy (3).
Reviewer 2 Report
- A brief summary
Schaafsma et al have a evaluated the association of immune infiltration and the specific expression of 23 immune related genes with survival in grades II-IV adult glioma using 3 separate clinical databases, TCGA, CGGA, and Rembrandt which revealed that these findings were mainly applicable to lower grade, II & III, gliomas and not so much in GBM.
- General concept comments
Strengths of the study: This is a well written and well executed study identifying the negative prognostic implications of monocyte/microglia infiltration and expression of specific immune related genes in glioma validated across 3 separate datasets. They also show the positive association of B and T lymphocytes in these tumors.
Weaknesses of the study: This study is very similar to one published in 2020, 10.21037/atm.2020.01.73, where the authors show a negative association between immune infiltration and certain immune signatures, including a 25 immune gene set. This gene set is different than the one identified in the reviewed manuscript although the algorithm for determining genes of interest appears to be very similar. The authors need to address the similarities and differences of their study with the previously published one and specifically discuss why different gene signatures were found to associate with risk of progression. The existence of a similar uncited study severely affects the impact of this otherwise well done study.
- Specific comments
In the introduction there are only references made to studies using the Bristol Myers Squibb agents, nivolumab and ipilimumab, with no mention of similar studies using pembrolizumab.
A recurring issue in the glioma literature is the way that TCGA organizes high and low grade glioma in its dataset. As the authors note, high grade gliomas include grades III & IV and low grade glioma, grades I & II. However, in the TCGA dataset “LGG” = grades II & III, and “HGG” = GBM, grade IV. Authors using this dataset need to specifically address this issue. I agree with the authors’ conclusions that the impact of the variables they evaluated was greater in the “LGG” data set than the “HGG” data set, however, it should be explicitly stated what these data sets entail so as not be clinically misleading. For example, the above cited paper describes them as “lower grade glioma”.
The use of microglia scores is a novel and important aspect of this study. However, it is confusing exactly what these parameters were. Some annotation suggests specific genes such as “Arg1” however others suggest gene set enrichment such as “Aging” or “Injury”. Please clarify what these designations signify in the methods. Similarly it is unclear what the phrase, “ When mouse microglia signatures used, human homologues used in calculation”. How did this work given that there were only 9 human microglia indicators and 20 mouse ones? Please clarify.
The authors found that increased immune gene signatures and expression of immune checkpoint molecules were actually associated with worse clinical outcomes which is different as they note from what is seen in other solid tumors. They hypothesize that this is due to monocyte/macrophage inhibition of an adaptive immune response as supported by their data. Is there further deconvolution data to support this? It would be extremely interesting to see which immune cell subsets might be expressing different genes. Are the infiltrating monocytes/macrophages clearly expressing “M2” markers? Or the authors found that both IFNgamma (immune activating) and TGFbeta (immune suppressive) signatures were both elevated in higher risk tumors. Since these pathways typically have opposing functions in the TME this should be further explored if possible. Understanding more about the infiltrating immune cells evaluated would increase the impact and novelty of this study.
Author Response
Dear Reviewer,
Please see our responses attached.

Reviewer 3 Report
Ref: Submission ID Cancers 1866932
Comments on Microglia-based gene expression signature highly associated with prognosis in glioma
The work done by Schaafsma et al. is a well written and authors tried to address relationship between immune genes over-expression and glioma patient prognosis where their data indicated a negative correlation i.e., overall decreased survival.
While the question targeted to be answered is innovative but as the authors already mentioned this is a preliminary in silico study with vital limitations which needs validation by further in vitro, in vivo, and pre-clinical studies to have a robust scientific fervor. The major concerns with the manuscript have been highlighted below needs to be thoroughly addressed before its publication in Cancers.
1. A graphical representation of the study especially showing the workflow and signaling molecules the authors have highlighted in their study should be included for better understanding of the study.
2. The title of the study is partially misleading and doesn’t support the findings of the study since its true for only LGG but not GBM. The title should be changed to “Microglia-based gene expression signature highly associated with prognosis in Low Grade glioma”
3. In the Introduction page 2, first line; please mention what are the other cell populations briefly?
4. Also, in the introduction section please introduce what are Nivolumab, bevacizumab and Ipilimumab with some more details.
5. The exact same sentence “Consequently, a better understanding of the malignant features of the TME in glioma is pertinent” is repeated twice in the abstract and introduction section but the authors report their findings on one of the non-malignant components or the immune cells. Provide the explanation for highlighting this sentence.
6. Figures should be prepared, keeping in mind the legibility and usefulness to corroborate the scientific findings. The Figure1 is of very poor quality, obscured with fonts difficult to decipher.
7. The authors stated that they estimated the abundance of six immune cell types commonly present in the TME: memory B cells, naïve B cells, CD4+ T cells, CD8+ T cells, NK cells and monocytes. But another very important cell type, promoting tumor formation by immunosuppressive functions the “Myeloid derived suppressor cells” which has multifaceted role in TME was not studied. These cells categorically are mostly pathologically activated monocytes and neutrophils. Authors should include the data from this cell type. Further, in context to this, were the monocytes studied here pathologically activated? The authors observed that when comparing overall survival, patients with high monocyte infiltration had significantly shorter survival as compared to patients with a broader immune cell infiltration pattern. Since, monocyte plays such an important role the authors should also look into the infiltration survival relationship of pathologically activated neutrophils.
8. After studying that IRG are negatively linked with glioma’s prognosis, the authors studied the patterns of infiltration and microglia’s abundance related studies in only LGG. Why did they not investigate these aspects in the GBM? Please clarify the reason. The authors should include the study about the patterns of infiltration on GBM in the revised manuscript. It is also highly recommended to study the microglia’s abundance in GBM and include in the revised manuscript.
9. The authors stated that “We indeed observed a difference among subtypes: astrocytomas contained the highest levels of microglia infiltration whereas oligodendrocytomas had the lowest levels of microglia infiltration”. However, on evaluating the overall survival, no difference in overall survival in astrocytomas was observed but patients with lower microglia infiltration had significantly longer survival in patients in oligoastrocytoma and oligodendrocytoma (Figure 4F)”. Please explain this very critical observation reported in the study.
10. The authors need to give a suitable explanation why they used mRNA data for studying the expression of immune checkpoints instead of the proteins while they seem to understand that expression level can only be only studied with proteins and not mRNA. In this context, they need to change the subheading “Microglia infiltration is associated with the expression of immune checkpoint genes and immuneregulatory pathways” and the subsequent figure legend for figure 5 by mentioning “mRNA level” instead of “expression” which is misleading.
Author Response
Dear Reviewer,
Please find our responses attached.

Round 2
Reviewer 2 Report
Thank you for the thorough revisions. Excellent work!
Reviewer 3 Report
Authors have mostly tried to address the major issues pointed out so its suitable to be published in its revised form